# Longitudinal change of cervical artificial disc motion following replacement

Jung Hyeon Moon[1], Chun Kee Chung[2,3,4,5]*, Chi Heon Kim[2,3,4], Chang-Hyun Lee[2], Sung Bae Park[6], Won Heo[7]

1 Department of Neurosurgery, Dongguk University Ilsan Hospital, Ilsan, South Korea, 2 Department of Neurosurgery, Seoul National University College of Medicine, Seoul, South Korea, 3 Neuroscience Research Institute, Seoul National University Medical Research Center, Seoul, South Korea, 4 Clinical Research Institute, Seoul National University Hospital, Seoul, South Korea, 5 Department of Brain and Cognitive Sciences, Seoul National University College of Natural Sciences, Seoul, South Korea, 6 Department of Neurosurgery, Seoul National University Boramae Medical Center, Seoul, South Korea, 7 Department of Neurosurgery, Gyeongsang Natinoal University Changwon Hospital, Changwon, South Korea

* chungc@snu.ac.kr

**Data Availability Statement:** All relevant data are in figshare: 10.6084/m9.figshare.11719842.

**Funding:** This research was supported by a grant of the Korea Health Technology R&D Project through the Korea Health Industry Development

## Abstract

We reviewed charts and radiologic studies of 30 patients operated upon by ADR with Mobi-C® in single level since 2006. All patients had healthy cervical facet joints (less than or equal to grade 1 according to grading systems for cervical facet joint degeneration) preoperatively. We assessed clinical outcomes with NDI and VAS on neck and arm over follow-up and also measured ROM at implanted segment on dynamic radiographs during follow-up. The mean follow-up period was 42.4 ± 15.9 months. We then assessed the linearity of changes in ROM at implanted segment through linear mixed model. All patients showed significantly improved clinical outcomes. ROMs at implanted segment were maintained at slightly increased levels until 24 months postoperatively (P = 0.529). However, after 24 months, ROMs at implanted segment decreased significantly until last follow-up (P = 0.001). In addition, the decreasing pattern after 24 months showed a regular regression (P = 0.001). This decline was correlated with decline of extension angle at implanted segment. Based on this regular regression, we estimated that ROMs at implanted segments would be less than 2 degrees at 10.24 years postoperatively. Even though implanted segment maintains its motion for some length of time, we could assume that an artificial disc would have limited life expectancy correlated with the decline of extension angle.

## Introduction

Anterior cervical discectomy and fusion (ACDF) is the gold standard for the treatment of degenerative cervical spine disease [1]. However, the long term results of ACDF have shown development of adjacent segment disease because of the loss of range of motion (ROM) at fused segments [2–5]. Therefore, cervical artificial disc replacement (ADR) has been suggested as an alternative to ACDF due to the preservation of mobility of implanted segments. There are numerous studies that have revealed the preservation of segmental ROM over follow–up without the development of adjacent segment disease after ADR [6, 7]. Cervical artificial disc replacement offers several theoretical and obvious advantages compared with ACDF.

Institute (KHIDI), funded by the Ministry of Health & Welfare, Republic of Korea (grant number: HC15C1320).

**Competing interests:** The authors have declared that no competing interests exist.

However, ADR also has problems such as heterotopic ossification or mechanical failure, which may raise concerns about the long-term fate of artificial discs [8]. Nevertheless, there are a number of papers that show how the ROM at the implanted segment changes with the passage of time after ADR [9].

The purpose of this study is to depict changing ROM patterns at the implanted segment over follow-up after ADR and to predict the life expectancy of artificial cervical disc. We performed this study in a cohort in which all patients had a healthy facet joint before surgery.

## Materials and methods

### Patient cohort and surgical technique

All data used in this study were approved by Institutional Review Board of Seoul National University Hospital. The Institutional Review Board (1610-104-801) approved our study. The IRB web address is https://cris.snuh.org. We reviewed the charts and radiological studies of 30 consecutive patients who were operated on using ADR at a single institute since 2006.

The patients had presented with radiating pain, paresthesia or weakness caused by cervical degenerative disease. We included patients who underwent ADR in a single level and excluded patients who underwent hybrid surgery (ADR and ACDF). We also excluded patients with trauma or tumors. All patients underwent ADR with Mobi-C prosthesis (LDR medical, France) in a single level. The mean follow-up period was 42.4 ± 15.9 months.

The Mobi-C, cervical artificial disc, is a semiconstrained mobile-bearing bone-sparing device. It is composed of two spinal plates consisting of cobalt, chromium, 29 molybdenum alloy (CoCrMo, ISO 5832–12) and an ultra-high-molecular-weight polyethylene (UHMWPE) mobile insert [10]. ADR was performed by 3 experienced surgeons at a single institute. The surgical technique consisted of a conventional anterior approach and discectomy followed by neural decompression. After decompression, the prosthesis was gently inserted into the disc space using a specific inserter. The primary anchoring optimization was obtained through compression with the Casper distractor. An X-ray (AP and lateral view) confirmed the adequate positioning of the implant. There were no differences in postoperative management among the 3 surgeons.

### Radiological assessment

Preoperatively, MRI, CT, and dynamic X-rays of the cervical spine were taken in all patients. Cervical facet joint degeneration was graded according to the literature [11, 12]. As shown in Table 1, cervical facet joint degeneration was classified into grades 0 to 4 according to presence/absence of osteophytes, hesubchondral sclerosis, and the irregularity of the apophyseal

**Table 1. Patient characteristics.**

| | |
|---|---|
| Number of patients (n) | 30 |
| Male: female (n) | 21: 9 |
| Mean age at surgery (years of age) | 44 |
| Mean follow-up length (months) | 42.4 |
| Grade for cervical facet joint degeneration (n) | Grade 0: 28 |
| | Grade 1: 2 |
| Implanted level (n) | C3-4: 1 |
| | C4-5: 6 |
| | C5-6: 15 |
| | C6-7: 8 |

joints. With the careful screening of the preoperative CT, we included only the patients who had healthy cervical facet joints (less than or equal to grade 1) and excluded patients with tumor or trauma. The follow-up dynamic X-rays were also taken in all patients. All patients were requested to flex and extend their necks to the extent they could tolerate for dynamic X-rays. Dynamic measurements with flexion and extension from a lateral view were subsequently taken at 3 months, 6 months, 9 months, 12 months, 2 years, 3 years, 4 years, and 5 years postoperatively.

We measured the flexion-extension ROM at the implanted segment on the lateral radiograph by a tangent method [13]. We also confirmed whether the implanted segment was fused or not by measuring the difference in interspinous processes on dynamic lateral radiographs. We considered an implanted segment to be fused if the difference in interspinous processes was below 2 mm on dynamic views [14]. The development of HO was assessed on lateral radiographs and was graded according to McAfee's criteria [15]. Two experienced observers measured all views. Because the values of the two observers were statistically consistent and significant, we performed an analysis with the median value of the two observers.

## Outcome assessments

Clinical outcomes were assessed with the neck disability index score (NDI) and with visual analog scales (VAS) for neck and arm pain. The NDI score was measured preoperatively and over follow-up. NDI success was defined as an improvement of scores greater than or equal to 15 points after surgery, which was used to evaluate a functional recovery. VAS for neck and arm pain was measured preoperatively and over follow-up. Neurologic status was also evaluated by the investigator through reflex test, motor and sensory function. Neurological success was defined as the absence of significant neurologic deterioration.

## Statistical analysis

To correct the intraobserver and interobserver reliability of the radiologic measurement, two experienced observers independently evaluated the radiographs of the patients. We then analyzed the values with a Bland–Altman plot to confirm a correspondence (Fig 1).

We used a linear mixed model to assess the longitudinal changes of ROMs at the implanted segments and compensated for missing values in our data. Additionally, we used regression analysis to assume when the implanted segment would lose its motion. We also used Kaplan–Meier curve analysis to analyze how many of the implanted segments maintained their motion during the follow-up period and when the risk of decreased ROMs at the implanted segment increased, compared to the normal segmental ROMs, which were based on a study by Lind et al. [16] (Table 2).

Statistical analysis was carried out using SPSS software for Windows (ver.21.0; SPSS Inc., Chicago, IL, USA). The results were considered as statistically significant at $p < 0.05$ (two-sided).

The demographics of the patients are shown in Table 1.

## Results

### Clinical results

All patients achieved an improvement in their symptoms. The NDI, on average, improved from 20.5 to 5.08 at the last follow-up, which represents a 74% improvement. Fifteen of the 30 patients achieved an improvement in their NDI scores higher than or equal to 15 points.

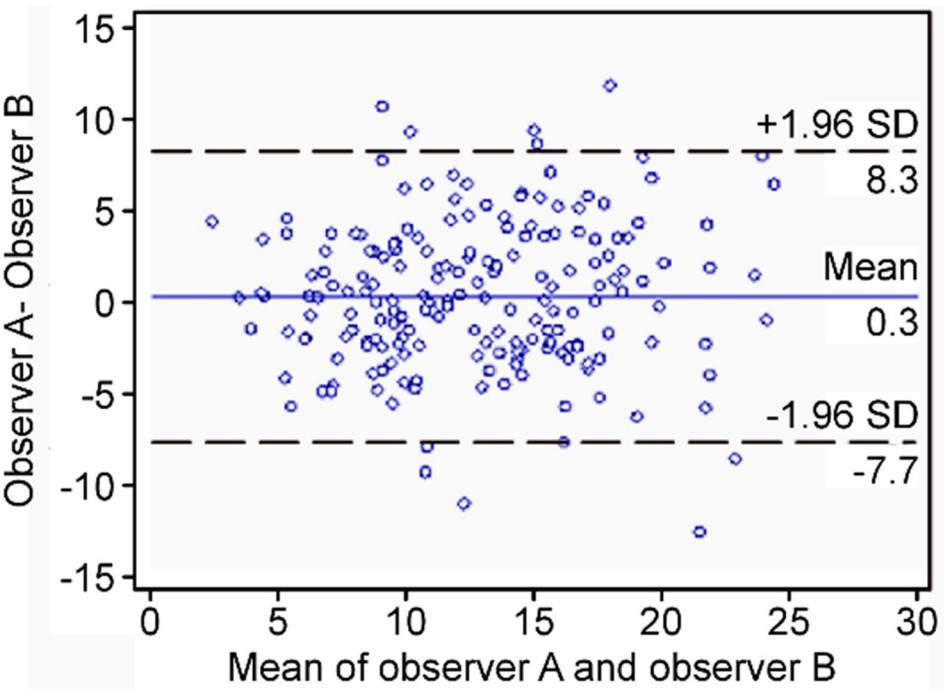

**Fig 1. Bland-Altman plot.** This statistical method is to confirm a correspondence.

The VAS score was reduced on average at each follow-up period. The neck VAS score on average reduced from 6.3 to 1.5 at the last follow-up, representing a 76% improvement, and the arm VAS score on average was also reduced from 6.9 to 0.5 at the last follow-up, representing a 92% improvement. All patients also achieved neurological success at the last follow-up.

There were no reoperations that can result from device failure or postoperative bleeding in our cohort.

## Radiographic results

The radiologic measurement was statistically correlated between the intraobserver and interobserver observations. Preoperatively, of the total 30 patients, 28 patients had grade 0 cervical facet joints; 2 patients had grade 1 cervical facet joints. This argues for the fact that most patients in our cohort had relatively healthy facet joints preoperatively [11, 12].

Heterotopic ossification (HO) was found in 13 of 30 patients (43%) at the last follow-up. Eight patients had an HO grade of 3, 4 patients had a grade of 2, and 1 patient had a grade of 1. There were no grade-4 HO patients in our cohort. Nonetheless, 3 of 30 patients were considered to lose their ROMs at the implanted segments at the last follow-up.

The longitudinal change of cervical artificial disc motion is shown in Fig 2A. We analyzed this longitudinal change with median values from two measurers. ROMs at the implanted

**Table 2. Normal cervical flexion and extension angles by Lind et al.**

|  | Number | Mean ± SD (˚) | | | | |
|---|---|---|---|---|---|---|
|  |  | C2-3 | C3-4 | C4-5 | C5-6 | C6-7 |
| Lind et al. [16] | 70 | 10 ± 4 | 14 ± 6 | 16 ± 6 | 15 ± 8 | 11 ± 7 |

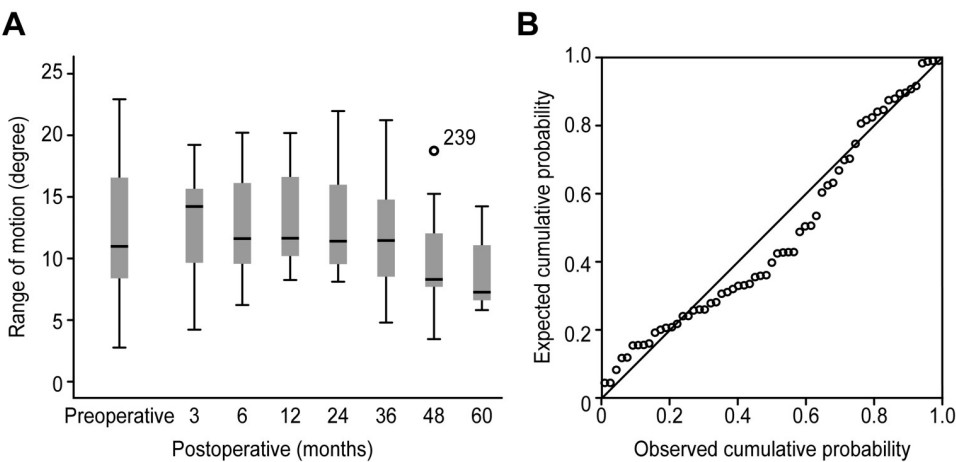

**Fig 2.** (A) Longitudinal changes of range of motion (ROM) at implanted segment. Longitudinal changes of range of motion (ROM) at implanted segment from preoperative to postoperative 60 months. This graph showed that ROM at implanted segment decreased significantly after 24 months. (B) Regular regression graph. The decreasing pattern of ROM at implanted segment after 24 months showed a regular regression.

segments did not change for 24 months, compared to the preoperative segmental ROMs ($P = 0.529$). However, ROMs at the implanted segments decreased significantly from 24 months to the last follow-up with regular regression ($p = 0.01$) (Fig 2B). Based on this regression, we could assume that cervical artificial disc would lose their function after 10.24 years postoperatively (less than 2˚) (Fig 3A). The linear mixed model revealed this trend.

Since we found that segmental ROMs decreased after 24 months, we analyzed how many of the implanted segments maintained their motion during the follow-up period compared to the normal segmental ROMs by using a Kaplan–Meier curve analysis. In the Kaplan–Meier curve analysis, the event was defined to be beyond a standard deviation of the normal ROM at each segment, which was based on a study by Lind et al. [16].

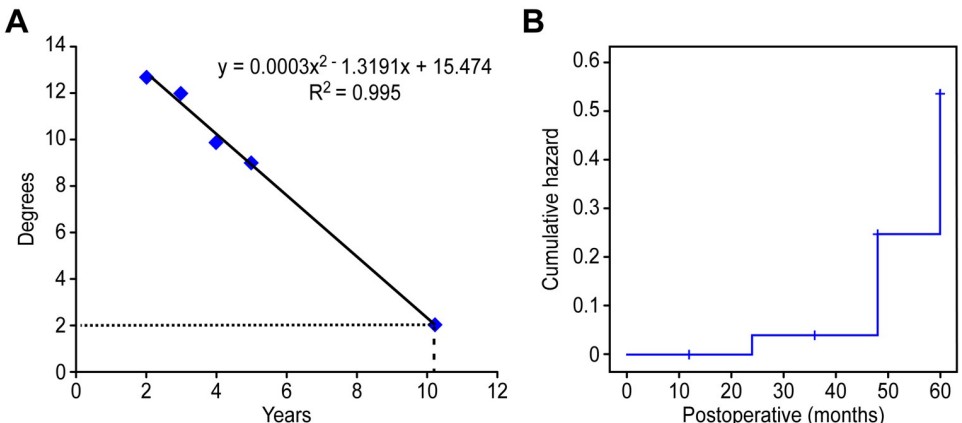

**Fig 3.** (A) The trend line. The trend line that showed when the range of motion (ROM) at implanted segment would be less than 2 degrees. This graph showed that ROM at implanted segment would be less than 2 degrees at 10.24 years postoperatively. (B) Hazard function. The event was defined to be beyond standard deviation of the normal ROM at each segment. This graph showed that the probability of a less than normal segmental ROM began to increase after 48 months.

In survival analysis, 80% of the implanted segments maintained their motion comparable to normal segmental ROMs until the last follow-up. However, a hazard function revealed that the probability of a less-than-normal segmental ROM began to increase after 48 months (Fig 3B).

We also analyzed separately the longitudinal change of flexion and the extension angle at the implanted segment. The flexion angle at the implanted segment (F-angle) decreased significantly at postoperative 3 months (P = 0.006) then, was maintained until the last follow-up (Fig 4A). However, the extension angle at the implanted segment (E-angle) increased significantly at postoperative 3 months (P = 0.001) then, and the angle was maintained until postoperative 24 months. After 24 months, E-angle decreased significantly until the last follow-up (P = 0.02) (Fig 4B). Based on each analysis, we assumed that the decline of E-angle after postoperative 24 months would influence the change of ROM at the implanted segment.

In analysis of cervical curvature, the cervical lordosis increased until postoperative 24 months then, was maintained until last follow-up (Fig 4C).

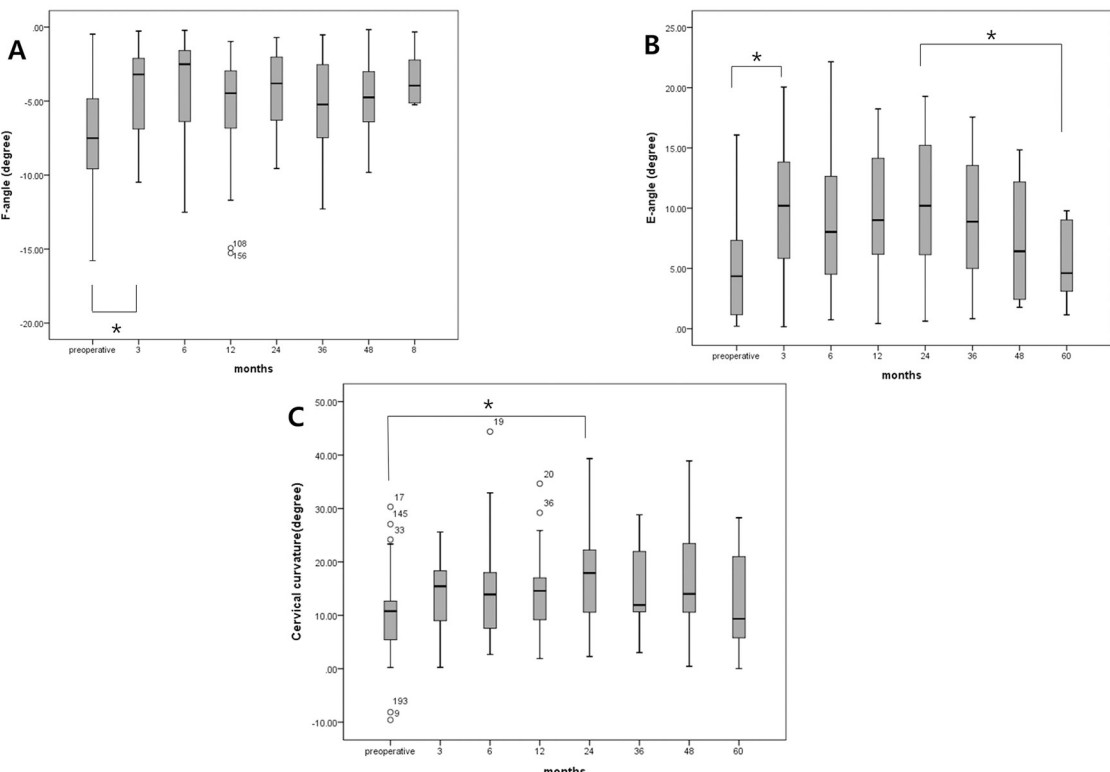

**Fig 4.** (A) Longitudinal changes of flexion angle at the implanted segment. Longitudinal changes of flexion angle at the implanted segment (F-angle) from preoperative to postoperative 60 months. This graph showed that F-angle decreased significantly at postoperative 3 months then, and the F-angle was maintained until the last follow-up. (B) Longitudinal changes of extension angle at the implanted segment. Longitudinal changes of extension angle at the implanted segment (E-angel) from preoperative to postoperative 60 months. This graph showed that E-angle increased significantly at postoperative 3 months then, was maintained until 24 months. However, after 24 months, E- angle decreased significantly until the last follow-up. (C) Longitudinal change of cervical curvature after artificial disc replacement. The cervical lordosis increased until postoperative 24 months then, and it was maintained until the last follow-up. * means statistically significant.

## Discussion

Even though ACDF is the gold standard for the treatment of degenerative cervical disease, many surgeons are searching for alternatives because of the likelihood of developing adjacent segment diseases following ACDF.

Because the loss of operated segmental motion caused adjacent segment diseases, ADR has been in the limelight as an alternative to ACDF.

There are numerous studies reporting that ADR is able to maintain segmental ROM at the implanted segment during follow-up [9, 17, 18]. However, there are also studies reporting that segmental ROM at the implanted segment tended to decrease with time [19–21].

To address this controversial issue, we performed a retrospective analysis to look at how the segmental ROMs at the implanted segments change during the follow-up period in a cohort that consisted of patients who had minimal facet degeneration and were operated upon with Mobi-C.

Since we tried to elucidate the change of ROM by only ADR, excluding the effect by facet joint degeneration, we included only the patients who had healthy cervical facet joints preoperatively [11, 12]. The incidence of heterotopic ossification, which is known as one of the major issues to cause the loss of motion after ADR, was 43%, which was similar to what has been reported in other studies [22, 23]. There were only 3 patients who lost their ROM at the implanted segment at the last follow-up. Nevertheless, there was a clear decreasing trend of ROMs at the implanted segments from postoperative 24 months.

As shown in Fig 2a, ROMs at the implanted segments were maintained until 24 months without a significant decrease (P = 0.529). After that, however, ROMs at the implanted segments started to decrease significantly until last follow-up (P<0.001).

Since we found that segmental ROMs decreased with time after 24 months, we analyzed how many of the implanted segments maintained their motion until the last follow-up, compared to segmental ROMs in a normal cohort. In survival analysis, 80% of the implanted segments were within the normal range, comparable to the segments in a normal cohort during follow-up. However, we found that the probability of less-than-segmental ROMs in a normal cohort began to increase sharply after 48 months (Fig 3B).

In addition, we noticed that the decreasing pattern of ROMs at the implanted segments after 24 months showed a regular regression (P = 0.01) (Fig 2B). We tried to predict when ROMs at the implanted segments would lose their motion (less than 2°) based on this regression. We found that ROMs at the implanted segments would lose their motion after 10.24 years postoperatively (Fig 3A).

There are numerous studies that advocate cervical artificial disc replacement due to its preservation of motion. These studies have tried to reinforce the power of evidence using multicenter studies [10, 19, 24]. Most of these studies insisted that an artificial disc replacement was superior to ACDF by comparing a preoperative segmental ROM with a postoperative ROM at the implanted segment, especially at the last follow-up. In other words, they explained how great a proportion of the implanted segments would maintain their motion at the last follow-up compared to a preoperative segmental ROM.

Although a multicenter analysis is a useful and powerful method for the limited-time comparison between pre- and post-operation, an analysis of a longitudinal trend in multicenter study is not easy. In the present study at a single center, we were able to analyze the long-term longitudinal change of ROM at implanted segments, even though it would not be prudent as a multicenter study. As a time point analysis, our results were similar to those of other multicenter studies because 80% of the implanted segments maintained their motion comparable to a normal segment until the last follow-up in our study. However, we analyzed the longitudinal

change of ROM at the implanted segments and thus confirmed that ROM at the implanted segments decreased significantly after 24 months postoperatively.

It is unclear whether ROM at the implanted segments will decrease continuously according to this pattern after 60 months or not. However, Putzier et al. reported that a Charité total disc replacement in the lumbar spine resulted in a high rate (60%) of spontaneous fusion or arthrodesis after an average follow-up of 17 years [25]. Although the kinetics of the cervical spine are different from those of the lumbar spine, we can assume that implanted segments with an artificial disc would lose their motion eventually.

In addition, there are also several studies showing a decreasing trend of segmental ROM after implantation even though ROM at the implanted segments were relatively preserved until the last follow-up [17, 18]. Burkus et al. [18] reported that ROM at the implanted segment were preserved until the last follow-up in their prospective randomly controlled study. They analyzed segmental ROMs at the implanted segment preoperatively and at 1.5 months, 3 months, 6 months, 12 months, 24 months, 36 months, and 60 months postoperatively. In their study, the mean ROMs at the implanted segments were 7.5˚ preoperatively, 7.3˚ at 36 months, and 6.4˚ at 60 months postoperatively. Although the implanted segmental ROM at 48 months postoperatively was not analyzed, it was clear that ROMs at the implanted segment decreased during the follow-up period, especially after 36 months. This result also shows that ROM at the implanted segments would decrease with time, even though this result did not perfectly coincide with our result.

## Limitations of the study

There were some limitations in the present study.

First, this study was a retrospective analysis at a single center. The selection bias and limited statistical power should be considered. Additionally, not all patients underwent serial radiographs during the follow-up period. Therefore, we tried to compensate the missing data with a statistical method (linear mixed model analysis). Prospective analysis was necessary to increase the reliability of our results. Second, a decreasing pattern does not mean the loss of segmental ROMs. It is hard to conclude when exactly an artificial disc loses its function. Therefore, it is necessary to study whether the degree of ROM should be maintained to prevent adjacent segment disease even though the ROM would not be within the range of a normal segment ROM. Third, we performed the study with a single device, the Mobi-C, which cannot represent all artificial disc devices.

Finally, a larger number of cases and an additional, longer follow-up period are necessary.

## Conclusion

We confirmed that the operated segmental ROMs began to decrease significantly after 24 months even though they were preserved until 24 months. Based on the linear regression of decreasing pattern, we could assume that the implanted segments would lose their motions someday correlating with the decrease of extension angle at implanted segment. Even though the implanted segment maintains its motion for some length of time, we could assume that an artificial disc would have a limited life expectancy correlated with the decline of extension angle.

## Author Contributions

**Conceptualization:** Chun Kee Chung.

**Data curation:** Jung Hyeon Moon, Chi Heon Kim, Chang-Hyun Lee, Won Heo.

**Formal analysis:** Jung Hyeon Moon, Chang-Hyun Lee, Won Heo.

**Investigation:** Jung Hyeon Moon, Won Heo.

**Methodology:** Jung Hyeon Moon, Chun Kee Chung, Chi Heon Kim, Chang-Hyun Lee.

**Resources:** Jung Hyeon Moon.

**Software:** Jung Hyeon Moon.

**Supervision:** Chun Kee Chung.

**Validation:** Jung Hyeon Moon, Chi Heon Kim.

**Visualization:** Jung Hyeon Moon.

**Writing – original draft:** Jung Hyeon Moon.

**Writing – review & editing:** Chun Kee Chung, Chi Heon Kim, Sung Bae Park.

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
