## [Decision Letter · Decision Letter 0]

4 Aug 2019

PONE-D-19-19220

Longitudinal change of cervical artificial disc motion following replacement

PLOS ONE

Dear Professor Chung,

Thank you for submitting your manuscript to PLOS ONE. After careful consideration, we feel that it has merit but does not fully meet PLOS ONE’s publication criteria as it currently stands. Therefore, we invite you to submit a revised version of the manuscript that addresses the points raised during the review process.

We would appreciate receiving your revised manuscript by Sep 18 2019 11:59PM. To enhance the reproducibility of your results, we recommend that if applicable you deposit your laboratory protocols in protocols.io, where a protocol can be assigned its own identifier (DOI) such that it can be cited independently in the future. For instructions see: http://journals.plos.org/plosone/s/submission-guidelines#loc-laboratory-protocols

We look forward to receiving your revised manuscript.

Kind regards,

Jonathan H Sherman

Academic Editor

PLOS ONE

Journal Requirements:

1. In ethics statement in the manuscript and in the online submission form, please provide additional information about the patient records used in your retrospective study. Specifically, please ensure that you have discussed whether all data were fully anonymized before you accessed them and/or whether the IRB or ethics committee waived the requirement for informed consent. If patients provided informed written consent to have data from their medical records used in research, please include this information.

2. Thank you for including your ethics statement in your methods section:

"The Institutional Review Board (1610-104-801) approved our study.".

i) Please amend your current ethics statement to include the full name of the ethics committee/institutional review board(s) that approved your specific study.

ii) Once you have amended this/these statement(s) in the Methods section of the manuscript, please add the same text to the “Ethics Statement” field of the submission form (via “Edit Submission”).

3. Thank you for including your funding statement; "No. The funders had no role in study design, data collection and analysis, decision to publish, or preparation of the manuscript."

Please provide an amended Funding Statement that declares *all* the funding or sources of support received during this specific study (whether external or internal to your organization) as detailed online in our guide for authors at http://journals.plos.org/plosone/s/submit-now.  

Please state what role the funders took in the study.  If any authors received a salary from any of your funders, please state which authors and which funder. If the funders had no role, please state: "The funders had no role in study design, data collection and analysis, decision to publish, or preparation of the manuscript."

4.

We note that you have indicated that data from this study are available upon request. PLOS only allows data to be available upon request if there are legal or ethical restrictions on sharing data publicly. For information on unacceptable data access restrictions, please see http://journals.plos.org/plosone/s/data-availability#loc-unacceptable-data-access-restrictions.

Reviewers' comments:

Reviewer's Responses to Questions

**Comments to the Author**

1. Is the manuscript technically sound, and do the data support the conclusions?

Reviewer #1: Yes

Reviewer #2: Yes

2. Has the statistical analysis been performed appropriately and rigorously? 

Reviewer #1: Yes

Reviewer #2: Yes

3. Have the authors made all data underlying the findings in their manuscript fully available?

Reviewer #1: Yes

Reviewer #2: Yes

4. Is the manuscript presented in an intelligible fashion and written in standard English?

Reviewer #1: Yes

Reviewer #2: Yes

5. Review Comments to the Author

Reviewer #1: The authors present a retrospective, single center, experience of 1 level cervical disc arthroplasties performed by 3 surgeons beginning in 2006 analyzing the progression of loss of ROM in the implanted segment over the course of the mean followup period at 42.4 months. The authors note the ROM was maintained until 24 months, after which, there was a statistically significant decline in the ROM. I would recommend the authors include data specifying the time intervals at which ROM was assessed to further characterize the rate of decline. The extrapolation of ROM decline after 24 months was performed with rigorous analysis so I feel this is appropriate and useful. Despite the small sample size, this study may provide valuable insight regarding an area severely lacking in data.

Reviewer #2: an important paper on the long term follow up of ROM for the Mobi-C prosthesis. It is retrospective in nature, but the methods are rigorous. Recommend acceptance, however, it is crucial to report, very clearly any conflicts of interests that the authors may have with regard to the implant.

6. PLOS authors have the option to publish the peer review history of their article (what does this mean?). If published, this will include your full peer review and any attached files.

Reviewer #1: No

Reviewer #2: Yes: Joseph OBrien, MD, MPH

---

## [Author Response · Author response to Decision Letter 0]

13 Oct 2019

Reviewer #1:

Thanks for your good comment. But we just followed up patients as scheduled time such as at 3 months, 6 months, 9 months, 12 months, 2 years, 3 years, 4 years, and 5 years postoperatively.

So we could not specify the time intervals at which ROM was assessed to further characterize the rate of decline. We are sorry about that.

Reviewer # 2:

Thanks for your good comment. We declare there are not any conflict of interest for this study.

All authors received no specific funding for this work

The database of this study was uploaded in Figshare. The web address is https://figshare.com/s/2f2f2f893c8bae1dd4ff

The DOI is 10.6084/m9.figshare.9975020

The email address for Non-author point of contact information with the SNU neurosurgery department is dayeon422@snu.ac.kr. She is a team Physician Assistant.

We saved all data as fully anonymized forms. IRB committee approved that there were not any ethical problems.

All authors received no specific funding for this work. Hence, the funders had no role in study design, data collection and analysis, decision to publish, or preparation of the manuscript.

---

## [Decision Letter · Decision Letter 1]

15 Nov 2019

PONE-D-19-19220R1

Longitudinal change of cervical artificial disc motion following replacement

PLOS ONE

Dear Professor Chung,

Thank you for submitting your manuscript to PLOS ONE. After careful consideration, we feel that it has merit but does not fully meet PLOS ONE’s publication criteria as it currently stands. Therefore, we invite you to submit a revised version of the manuscript that addresses the points raised during the review process.

We would appreciate receiving your revised manuscript by Dec 30 2019 11:59PM. To enhance the reproducibility of your results, we recommend that if applicable you deposit your laboratory protocols in protocols.io, where a protocol can be assigned its own identifier (DOI) such that it can be cited independently in the future. For instructions see: http://journals.plos.org/plosone/s/submission-guidelines#loc-laboratory-protocols

We look forward to receiving your revised manuscript.

Kind regards,

Jonathan H Sherman

Academic Editor

PLOS ONE

Reviewers' comments:

Reviewer's Responses to Questions

**Comments to the Author**

1. If the authors have adequately addressed your comments raised in a previous round of review and you feel that this manuscript is now acceptable for publication, you may indicate that here to bypass the “Comments to the Author” section, enter your conflict of interest statement in the “Confidential to Editor” section, and submit your "Accept" recommendation.

Reviewer #1: All comments have been addressed

Reviewer #3: All comments have been addressed

2. Is the manuscript technically sound, and do the data support the conclusions?

Reviewer #1: Yes

Reviewer #3: Yes

3. Has the statistical analysis been performed appropriately and rigorously? 

Reviewer #1: Yes

Reviewer #3: Yes

4. Have the authors made all data underlying the findings in their manuscript fully available?

Reviewer #1: Yes

Reviewer #3: Yes

5. Is the manuscript presented in an intelligible fashion and written in standard English?

Reviewer #1: Yes

Reviewer #3: Yes

6. Review Comments to the Author

Reviewer #1: The authors have addressed the reviewers suggestions and produced a final manuscript adequate for publishing.

Reviewer #3: The authors present a retrospective study aimed to evaluate and to depict changing ROM patterns at the implanted segment over follow-up 1 after ADR and to predict the life expectancy of artificial cervical disc. The authors evaluated data from 30 patients after single level ADR at a single institution from 2006 to up to 5 years post operatively.

7. PLOS authors have the option to publish the peer review history of their article (what does this mean?). If published, this will include your full peer review and any attached files.

Reviewer #1: No

Reviewer #3: No

---

## [Author Response · Author response to Decision Letter 1]

20 Dec 2019

Reviewer #1:

Thanks for your good comment. But we just followed up patients as scheduled time such as at 3 months, 6 months, 9 months, 12 months, 2 years, 3 years, 4 years, and 5 years postoperatively.

So we could not specify the time intervals at which ROM was assessed to further characterize the rate of decline. We sorry about that.

Reviewer # 2:

Thanks for your good comment. We declare there are not any conflict of interest for this study.

All authors received no specific funding for this work

Reviewer # 3:

#1

Two observers in this study are spine surgeons. The first observer is Jung Hyeon Moon, M.D., first author, and the other observer is Won Heo, M.D., last co-author.

The each authors work as clinical professor in a different medical center at a different region.

They reached an agreement of in common about using tangent method for measurement.

The reliability of the radiologic measurements from two observers was confirmed with a Bland-Altman plot that is statistical method to confirm a correspondence.

We described these in materials and methods.

#2

All included patients had degenerative changes and correlating symptoms at the operated level. 

For confirming a pure motion of artificial disc, we only included the patients who had healthy cervical facet joint (less than or equal to facet joint degeneration grade 1) with the careful screening of the preoperative CT. we described these in radiological assessment.

#3

We could not compare outcomes by cervical level within the cohort. Because the number of the operated levels were not even in this study. 

For example, the number of patients who were operated at C5-6 level were 15, whereas the number of patient who was operated at c3-4 level was only 1.

We described these patient characteristics on Table 1.

#4

We tried to correct some grammatical errors. We highlighted changes in revised manuscript.

---

## [Decision Letter · Decision Letter 2]

22 Jan 2020

Longitudinal change of cervical artificial disc motion following replacement

PONE-D-19-19220R2

Dear Dr. Chung,

We are pleased to inform you that your manuscript has been judged scientifically suitable for publication and will be formally accepted for publication once it complies with all outstanding technical requirements.

With kind regards,

Jonathan H Sherman

Academic Editor

PLOS ONE

Additional Editor Comments (optional):

Reviewers' comments:

Reviewer's Responses to Questions

**Comments to the Author**

1. If the authors have adequately addressed your comments raised in a previous round of review and you feel that this manuscript is now acceptable for publication, you may indicate that here to bypass the “Comments to the Author” section, enter your conflict of interest statement in the “Confidential to Editor” section, and submit your "Accept" recommendation.

Reviewer #3: All comments have been addressed

2. Is the manuscript technically sound, and do the data support the conclusions?

Reviewer #3: Yes

3. Has the statistical analysis been performed appropriately and rigorously? 

Reviewer #3: Yes

4. Have the authors made all data underlying the findings in their manuscript fully available?

Reviewer #3: (No Response)

5. Is the manuscript presented in an intelligible fashion and written in standard English?

Reviewer #3: Yes

6. Review Comments to the Author

Reviewer #3: The authors addressed the comments appropriately in their revision. The manuscript is sound and the data is available.

7. PLOS authors have the option to publish the peer review history of their article (what does this mean?). If published, this will include your full peer review and any attached files.

Reviewer #3: No

---

## [Editor Report · Acceptance letter]

5 Feb 2020

PONE-D-19-19220R2 

Longitudinal change of cervical artificial disc motion following replacement 

Dear Dr. Chung:

I am pleased to inform you that your manuscript has been deemed suitable for publication in PLOS ONE. Congratulations! Your manuscript is now with our production department. 

With kind regards,

on behalf of

Dr. Jonathan H Sherman 

Academic Editor

PLOS ONE